# Chromosome Doubling in Genetically Diverse Bilberry (*Vaccinium myrtillus* L.) Accessions and Evaluation of Tetraploids in Terms of Phenotype and Ability to Cross with Highbush Blueberry (*V. corymbosum* L.)

Malgorzata Podwyszynska [1],*, Katarzyna Mynett [1], Monika Markiewicz [1], Stanisław Pluta [2] and Agnieszka Marasek-Ciolakowska [1]

[1] Department of Applied Biology, The National Institute of Horticultural Research, Konstytucji 3 Maja 1/3 Str., 96-100 Skierniewice, Poland; katarzyna.mynett@inhort.pl (K.M.); monika.markiewicz@inhort.pl (M.M.); agnieszka.marasek@inhort.pl (A.M.-C.)

[2] Department of Horticultural Crop Breeding, The National Institute of Horticultural Research, Konstytucji 3 Maja 1/3 Str., 96-100 Skierniewice, Poland; stanislaw.pluta@inhort.pl

* Correspondence: malgorzata.podwyszynska@inhort.pl

**Abstract:** To expand the gene pool and introduce new traits to the tetraploid cultivars of *Vaccinium corymbosum* from wild diploid species *V. myrtillus*, it is necessary to double the chromosome number in diploid species in order to overcome a post zygotic crossing barrier and a strong triploid block, existing within the genus *Vaccinium*. Five genetically diverse bilberry genotypes were selected from 21 accessions taken from the breeding collection of the National Institute of Horticultural Research (Skierniewice, Poland) for this study. The bilberry genotypes were derived from the Polish locations of Bolimów Landscape Park, Budy Grabskie and forest complex Zwierzyniec (Łódź Province), and habitats in Norway. The selection of genotypes was made based on the analysis of amplified fragment length polymorphism (AFLP-PCR). Analysis of the Jaccard similarity indexes and the UPGMA method revealed that the examined accessions formed two main groups on the dendrogram. The first group consisted of accessions from Norway, while the second group agglomerated Polish accessions. A further two classes were distinguished in the Polish group: the first included accessions from Budy Grabskie and the second from Zwierzyniec, located ca. 9 km from Budy Grabskie. In order to obtain plant material for in vitro polyploidisation, in vitro shoot cultures of the selected accessions were initiated and multiplied. Both antimitotics used, colchicine and APM, induced tetraploids for all of the accessions. The obtained tetraploids were multiplied, rooted ex vitro and grown in a greenhouse and then in a field. The first flowering was observed in 1.5-year-old plants, either diploid or tetraploid. Diploids bloomed slightly earlier and more profusely than tetraploid plants. Compared to diploids, autotetraploids had significantly larger flowers by ca. 64% and larger pollen tetrads by ca. 35%. The germination capacity of pollen tetrads was high in tetraploids (87.8%), although slightly lower than in diploids (94.3%). After pollinating the flowers of three highbush blueberry cultivars with pollen from the bilberry tetraploid accession, J-4-4x, the plants formed fruits, some of which contained properly formed seeds. The effectiveness of interspecific crossing between *V. corymbosum* and tetraploid *V. myrtillus*, defined as the percentage of obtained seedlings in relation to the number of pollinated flowers, was highest (53.3%) in the blueberry 'Liberty', and lower in 'Bluecrop' and 'Northland', 14.8% and 10.0%, respectively. Before using the seedlings for further breeding, their hybridity will be confirmed by molecular markers and the phenotype will be evaluated.

**Keywords:** AFLP; polyploidisation; colchicine; amiprophos methyl; genetic variation; in vitro culture; *Vaccinium myrtillus*

## 1. Introduction

The fruits of bilberry (*Vaccinium myrtillus* L.) are one of the richest sources of biologically active compounds, mainly phenolics, including anthocyanins. Their total content ranges from 300 to 1200 mg/100 g of F.W., which is much higher than in other *Vaccinium* species [1–3]. For comparison, the fruits of highbush blueberry cultivars commonly grown commercially in Poland and throughout the world contain, on average, about 200 mg/100 g F.W. On the other hand, the cultivation of highbush blueberry (*Vaccinium corymbosum* L.) shows the fastest growing trend in fruit production in Poland [4]. Poland has recently produced approximately 38,1000 tons of these fruits, which ranks it second in Europe and eighth in the world [5]. Rapid changes in agricultural technology and growing interest in the consumption of blueberry fruits due to their prohealth benefits have directed breeders' attention to developing improved cultivars. However, highbush blueberry cultivars with purple fruit flesh, containing high anthocyanin content, are not available yet. The fruit flesh of cultivated Vaccinium species is translucent (colourless), as it does not contain anthocyanins. In contrast, the bilberry (*V. myrtillus*) is characterised by purple fruit flesh. Bilberry is a native wild-growing shrub which occurs in acidic and humid forest soils in Europe, Asia and North America [2,6]. Obtaining highbush blueberry cultivars with anthocyanin-containing fruit flesh would significantly increase their value related to human health and would create the potential for the use of such fruits in the food and pharmaceutical industries. The introduction of the trait of anthocyanin-containing fruit flesh into blueberry is theoretically possible by interspecific highbush blueberry and bilberry hybridisation and further selection. However, due to the crossbreeding barrier, such hybrids have not been obtained thus far.

The genus *Vaccinium* belongs to the heather family (Ericaceae) and consists of two subgenera (*Vaccinium* and *Oxycoccus*) with about 150–450 species divided into 35 sections [6,7]. In each of the sections of high significance to breeding (*Myrtillus*, *Cyanococcus* and *Oxycoccus*), there are genotypes with different ploidy levels (2x, 4x and 6x). Bilberry (*V. myrtillus*) is a diploid (2n = 2x = 24) and belongs to the Myrtillus section. Among the cultivated species, the most important species in Poland and across the world are North American species belonging to the *Cyanococcus* section: highbush blueberry (4x) (*V. crymbosum*); of lesser importance—lowbush blueberry (4x) (*V. angustifolium* L.); their hybrids half-high blueberry (4x) (*V. angustifolium* × *V. corymbosum*) and *V. darrowii* (2x); rabbiteye blueberry (6x) (*V. virgatum*, syn. *V. ashei*). Other wild species, mainly diploid, also play a role in breeding. However, in the genus *Vaccinium*, obtaining ineploid hybrids, e.g., between diploids and tetraploids, is very difficult due to the partial triploid block [8–10]. The partial triploid block is a postzygotic incongruity manifested probably as the cellularisation failure of the endosperm, which is a tissue with a particular sensitivity to the imbalance between maternal and paternal genomes derived from parents differing in ploidy level, as shown in the extensive studies on interploid hybridisation of *Arabidopsis* sp. [11]. As a result, the endosperm does not develop and embryo formation is inhibited. To bypass this obstacle of interploid hybridization, [11] generated autotetraploids for a wild diploid *Arabidopsis lyrata* species, which were then crossed with natural tetraploid *A. arenosa*, which resulted in fertile hybrids. A similar mechanism to the postzygotic hybridisation barrier exists probably in the genus *Vaccinium*. Thus, obtaining triploid hybrid seeds is difficult or impossible, as recently reported by Norden et al. [9] and Cabezas et al. [10]. To expand the gene pool and introduce new traits to the tetraploid cultivars of *V. corymbosum* from wild diploid species (e.g., a very desirable trait of the presence of anthocyanin in the fruit flesh), it seems useful to double the chromosome number in diploid species obtaining autotetraploids [7]. The formation of autotetraploids within the *Vaccinium* species was induced by soaking seeds in colchicine solutions, which is, however, a low-efficiency method. Mitotic tetraploids were obtained only for a few diploid species [8,12,13]. These authors then obtained hybrids between autotetraploids of several wild species and *V. corymbosum*. However, up to now, no hybrids between bilberry and highbush blueberry have been obtained.

In our previous studies, we developed methods for the in vitro polyploidisation of daylily [14], apple [15], tulip [16] and blackcurrant [17]. In those methods, the shoots were treated with antimitotic agents such as colchicine, oryzalin, trifluralin and amiprophos methyl (APM). Our studies on these plants showed large differences between genotypes of particular species in their response to antimitotic agents. Therefore, it is necessary to optimize, for individual genotypes, the polyploidisation conditions such as treatments with various antimitotic agents at different concentrations. In addition, the polyploidisation process itself is an important source of variation [18]. The multiplication of the same genes in newly formed polyploids often translates into the intensification of the quantitative traits compared to their diploid ancestors. As a result of genetic changes, newly formed polyploid genotypes are often characterised by better quality of morphological features: vigorous growth; larger organ sizes, e.g., flowers, fruits; often higher chlorophyll content; fewer flowers in the inflorescence; sometimes shorter shoots; a more dense plant habit. Although tetraploids are usually fertile, pollen germination frequency is much lower compared to diploid forms [14,19]. Consequently, newly obtained autotetraploids are always assessed for phenotype and suitability for further breeding.

In nature, bilberry reproduces mainly vegetatively through underground rhizomes of sympodial branching, forming concentric, genetically homogeneous clusters of plants, reaching up to 15 m in diameter, with individual shrubs living up to 34 years [6]. *V. myrtillus* is partially self-pollinating, which leads to inbred depression [20] and could be associated with low genetic variability. The bilberry breeding collection at the Department of Horticultural Crop Breeding, the National Institute of Horticultural Research, Skierniewice, Poland (NIHR) consists of 21 accessions collected at Polish locations (the Bolimów Landscape Park (Łódź Province) and the forest complex Zwierzyniec close to Skierniewice (central Poland)) and in boreal forests in Norway. However, the genetic variability between the accessions collected was not known and it was necessary to evaluate it using any molecular markers. In our study, we decided to use amplified fragment length polymorphism (AFLP) as it is a relatively inexpensive DNA analysis of high reproducibility, resolution and sensitivity, having the capability of the identification of genetic variation within closely related genotypes [21–23]. Various molecular markers have been used for identification of *Vaccinium* species, clones, cultivars and genetic population structure for over 20 years. Random amplified polymorphic DNA (RAPD) markers discriminated the three *Vaccinium* species, namely cranberry, lowbush blueberry and lingonberry [24]. Using RAPD markers, a genetic variability was evaluated in *V. uliginosum* [25] and a collection of wild cranberry (*V. macrocarpon*) clones [26]. In turn, inter simple sequence repeats (ISSR)-based analysis was applied for the genetic variability evaluation of lingonberry (*V. vitisidaea*) [27] and lowbush blueberry (*V. angustifolium*) clones [28]. Debnath [29] evaluated the diversity of several wild blueberry clones and cultivars using EST-PCR (expressed sequence tags-PCR) and EST-SSR (simple sequence repeats microsatellites or genic microsatellites) markers. Recently, Alam et al. [30] used genotyping-by-sequencing (GBS) to analyse the genetic variation in 56 lingonberry samples collected from various locations in Canada.

For research purposes, bilberry has been propagated in vitro, although there are only a few reports on this [31–33]. In contrast, much more research has been done on the micropropagation of other *Vaccinium* species [34], especially *V. corymbosum* [35,36]. Most often, *Vaccinium* species are propagated on Anderson's medium [37] in the presence of zeatin or isopentenyladenine (iP), and shoots are rooted in vitro on media containing indole-3-acetic acid (IAA) or indole-3-butyric acid (IBA). In our studies, we used an in vitro propagation method developed by Litwińczuk [36] for highbush blueberry.

The aim of our research was to select genetically diverse bilberry accessions, subjecting them to polyploidisation in order to obtain autotetraploids capable of crossing with cultivated species of highbush blueberry.

## 2. Materials and Methods

### 2.1. Plant Material

Five genetically diverse bilberry genotypes, selected from 21 accessions maintained in the NIHR breeding collection in Skierniewice, Poland, were used for the research. Genotypes were derived from Polish habitats of the Bolimów Landscape Park (Budy Grabskie; 52.0019.70 N, 20.1128.11 E) and forest complex Zwierzyniec (Skierniewice; 51.9418.7 N, 20.05246 E), both in Łódź Province and from a Norwegian location: Tromsø, Holt, 9269 (69.3856.04 N, 18.5718.29 E) and Ås, Akershus (Norwegian Institute of Bioeconomy Research, kindly provided by Dr. Rolf Nestby).

The selection of genotypes for polyploidisation was made based on the analysis of genetic differences between bilberry accessions using AFLP markers.

In a preliminary study, in 2018, in vitro shoot cultures were initiated and polyploidisation was done for the bilberry accession J-4. The J-4 tetraploids were then rooted, cultivated ex vitro during 2019–2021 and, finally, their phenotype was observed in 2021. In the case of other genotypes: J-3, J-5, J-8 and J-9, in vitro shoot cultures were established at the beginning of 2021 and, soon, the obtained shoots were polyploidised. In vitro cultures and polyploidisation were performed for all five genotypes according to the same procedures described below.

### 2.2. Selection of Bilberry Accessions Based on AFLP Analysis

Genetic characterisation of bilberry genotypes was made using AFLP analysis. AFLP fingerprinting was performed according to Vos [38] with 10 AFLP primer pairs [39,40]. Genomic DNA was extracted from fresh bilberry leaves using the DNeasy Plant Mini Kit (Qiagen, Hilden, Germany) in three replicates for each sample tested. The concentration and purity of the DNA were determined using an Epoch spectrophotometer (BioTek Instruments, Winooski, VT, USA). The genomic DNA (50 ng) was digested with *Pst*I and *Mse*I endonucleases (Thermo Fisher Scientific, Waltham, MA, USA) and ligated with appropriate adapters, specific for the restriction sites. Amplification of obtained DNA fragments was performed using a T100$^{TM}$ thermal cycler (Bio-Rad, Hercules, CA, USA) in two steps: preselective amplification with primers complementary to adapters sequences and selective amplification using primers with two or three additional nucleotides at the 3' end. Products of selective PCR were separated on 6% denaturing polyacrylamide gel through electrophoresis on Dual Dedicated Height Nucleic Acid Sequencer (C.B.S. Scientific, Del Mar, CA, USA). The separated AFLP products were silver stained in a silver nitrate solution and the gel was dried, described and photographed. The size of the amplicons was assessed against size standards: 10 bp DNA Ladder (Thermo Fisher Scientific, Waltham, MA, USA) and 50 bp DNA Ladder (Thermo Fisher Scientific, Waltham, MA, USA). Amplicons generated in AFLP analysis were scored manually as present (1) or absent (0). Only bright and reproducible amplicons were scored. During the analysis of electrophoregrams, the number of AFLP products, their size and their diversity between tested genotypes were evaluated. On the basis of the obtained data, the Jaccard similarity indexes were calculated and an UPGMA (unweighted pair group method with arithmetic average) dendrogram was prepared using the D-UPGMA software (URV, Tarragona, Spain). Five bilberry accessions showing vigorous growth and higher genetic diversity were selected for further study.

### 2.3. In Vitro Shoot Culture Initiation and Shoot Micropropagation

In order to obtain plant material for in vitro chromosome doubling, the in vitro shoot cultures of bilberry were initiated for genotypes selected on the basis of AFLP analysis. In vitro shoot culture initiation and shoot multiplication were performed according to the method reported for highbush blueberry by Litwińczuk [36]. Briefly, as initial explants, axillary buds of plants growing in a plastic tunnel were used. Buds with a fragment of shoot were disinfested in standard procedure and placed on the medium for an initial stage at 20 °C in darkness for the first 7–10 days, then at 16-h standard photoperiod (PPFD of

30 µmol m$^{-2}$ s$^{-1}$, 18 W, 2000 lm, 2700–3200 K—Spectrum LED lamps, Schneider Electric, Warsaw, Poland). After six weeks, the developing shoots were subcultured onto fresh medium. The modified Anderson [37] medium supplemented with iP and IBA were used for the initial stage [36]. Then, the developing shoots were transferred to the multiplication medium with a similar composition.

### 2.4. Optimisation of In Vitro Polyploidisation Method

The conditions for treatments of shoots with antimitotic agents were optimised. Shoots (explants) 2–3 cm in length derived from 8-week multiplication subculture were treated with antimitotic agents by incubation for seven days in darkness in the multiplication medium containing colchicine (125 and 250 mg L$^{-1}$) or APM (5 and 10 mg L$^{-1}$). Colchicine was dissolved in distilled water and filtered through microporous filter membrane of 0.22 µm (Nalgene$^{TM}$ Sterile Syringe Filters, Thermo Fisher Scientific, Waltham, MA, USA), and APM was dissolved in 99.8% ethanol and directly added to the medium. The antimitotics were included in the media after autoclaving. Shoots were placed on the medium horizontally. After antimitotic treatment, shoots were transplanted to a similar medium that did not contain antimitotics and that was maintained in darkness for four weeks and then in standard photoperiod for an additional four weeks. The shoots were subcultured every eight weeks on multiplication medium with 16-h photoperiod with PPFD as described above. The phytotoxicity of antimitotic agents was assessed 8 weeks after antimitotic treatments; the observations concerned the survival frequency of the shoot explants and the number of regenerated shoots per explant.

For each treatment, 5 Erlenmeyer flasks with 6 shoots were used. The flasks were placed randomly on the shelf in the growth room.

### 2.5. Detection of Bilberry Tetraploids Based on Flow Cytometry

Tetraploids were detected using a flow cytometry (FCM) analysis. Samples (leafy shoot fragments, ca. 1 cm long) taken 16 weeks after antimitotic treatments from regenerated shoots were used for analysis. The nuclei were extracted from the chopped plant material in a Petri dish that contained 0.5 mL nuclei isolation Partec buffer [41] with 1% polyvinylpyrrolidone (PVP) and to which the fluorescent stain 4′,6-diamidino-2-phenylindole (DAPI) (50 µg mL$^{-1}$) was added. After adding 1 mL of the isolation buffer, the samples were filtered through a 30-µm filter and incubated at room temperature for 60–70 min in darkness. The fluorescence of the nuclei was measured using a CyFlow Ploidy Analyser (Partec, Germany), with UV-LED 365 nm. Data were analysed by means of CyView software (CyFlow PA, Partec, Germany). Samples with at least 2000 nuclei were measured. As an external standard, shoot samples of wild diploid *V. myrtillus* from in vitro cultures were used.

### 2.6. In Vitro Rooting Shoots of Selected Tetraploids and Their Diploid Counterparts, Acclimatisation to Ex Vitro Condition and Further Cultivation

The clumps of 3–5 shoots derived from 12-week subculture of diploid (untreated with antimitotics) and autotetraploid J-4 clones 4x-1, 4x-2, 4x-5 and 4x-7 were rooted directly in a mixture of peat, sand and sawdust (1:1:1, *v/v/v*) in a propagation tray (multiplates 3 × 3 cm cells). During the rooting and acclimatisation, plants were grown in plastic mini-greenhouses with two adjustable dial ventilators, in a growth room at a temperature of 23 °C under LED light (PPFD 60 µmol m$^{-2}$ s$^{-1}$), in October 2019. Three months later, mini-greenhouses with plants were transferred to a greenhouse and cultured at 20 °C with supplementary light provided by sodium lamps for 12 h. Two weeks later, plants were transplanted to 6 × 6 cm plastic pots with the same substrate. Plants were fertilised every 3–4 weeks with fertiliser for blueberry at a concentration of 1–2 g L$^{-1}$ (INCO S.A., Góra Kalwaria, Poland). In May 2020, plants were transplanted in 1 L pots and cultivated outdoors under shade cloth, fertilised as described above and watered as needed. The plants were overwintered outdoors (covered with 20 cm straw), then cultivated during the second season in 2021, similarly to the first season.

We obtained autotetraploids of J-4 bilberry during the first series of polyploidisation in 2019 (in preliminary study). Therefore, we had enough older plants that produced flowers. Consequently, it was possible to observe them in the generative phase and use them for crossing.

### 2.7. Phenotypic Evaluation

In order to investigate the differences between the obtained J-4 tetraploids and their diploid counterparts, the phenotypic evaluation of plants was performed. Observations at flowering of 18-month-old plants were done in April/May 2021. Evaluation of some morphological parameters of two-year-old plants were performed in September.

The evaluation of the following plant growth parameters and morphological traits were performed: the length and diameter of shoots; leaf size using a planimeter (Area Metr AM350, ADC BioScientific Ltd., Hoddesdon, UK); relative chlorophyll content measured with a CCM-200 Chlorophyll Content Meter (Opti-Sciences Int., Hud-son, NH, USA); stomata size and density using a method of Dyki and Habdas [42]. Ten leaves of each genotype were used for measurements. Stomata measurements were determined for five leaves ($\times$10 stomata) per genotype using a Nikon Eclipse 80i microscope (Nikon, Tokyo, Japan) with the program NIS-Elements BR 2.30 at 400$\times$ magnification; stomata density was calculated from five leaves $\times$10 fields of view at 400 times magnification. Microscopic photographic documentation was made using a CCD PS-Fi1 monochrome digital camera (Nikon, Tokyo, Japan).

Evaluations at flowering were performed for the following traits: flowering time, flower size (5–10 flowers of each genotype), pollen size and viability. Four to six plants of the selected tetraploid clones were assessed (depending on the number of obtained plants). A mixed sample of pollen from 3–5 flowers taken from plants of each genotype were stained with Alexander's stain [43]. The microscopic measurements of pollen grain length were done as described for stomata measurements. The pollen germination was estimated on microscope slides with sucrose solution of 15% after 6 h of incubation at room temperature. 100 pollen grains were observed for each genotype.

### 2.8. Hybridisation

In order to correlate the flowering time of both species, maternal (seed) plants of high-bush blueberry cultivars ('Bluecrop', 'Northland' and 'Liberty') were grown in a plastic tunnel and bilberry plants of tetraploid clone J-4-4x-4 as pollen parent were grown in a greenhouse from early April 2021 (plants were transferred to the greenhouse from outdoors).

Freshly opened bilberry flower buds were collected, and stamens remaining on the flower after the corolla was removed were dried. Pollen (remaining in the tetrads) released from the anthers was applied manually to the stigmas of the pistils of the emasculated flower buds of maternal highbush blueberry plants. Cross-fertilised flowers were labelled and counted and the shoot was immediately covered with nylon mesh to protect against insects. In July, fruits were collected, and seeds were extracted from them and dried for one week, then sown into peat substrate as used for growing young bilberry microplants. The numbers of fruits (matured and green) and properly formed seeds were counted for each cross-combination. Seedlings were grown and fertilised for the first 8 weeks in mini-greenhouses as described above for rooted micropropagated bilberry plants. The number of germinating seedlings was counted for each cross-combination.

### 2.9. Statistical Analysis

Data concerning the phytotoxicity of antimitotic agents and phenotype evaluation were subjected to one-way analysis of variance (ANOVA). The mean values were compared using Duncan's test at $p < 0.05$. Statistical analyses were performed using the STATISTICA program (StatSoft v. 13.1).

## 3. Results

### 3.1. Selection of Bilberry Accessions Based on AFLP Analysis

The total number of amplification DNA products (amplicons) obtained in the presence of 10 AFLP primer pairs was 924, of which 204 (22%) were polymorphic (Table S1). The lengths of the evaluated AFLP amplicons ranged between 50 and 900 base pairs. On average, there were 92.4 amplicons per pair of primers, and 44 amplicons per one accession tested. The number of amplicons generated by one primer pair ranged from 62 to 148. On average, there were 20.4 polymorphic amplicons for a primer pair and 9.7 for a single accession. AFLP primers generated from 3 (3%) to 50 (47%) polymorphic products (Figure S1).

The mean value of the genetic similarity indices of the accessions was high and amounted to 0.909, ranging from 1.000 (between accessions T12 and T13 from the Polish habitat Grabskie Budy) and 0.992 (accessions T19 and T20, as well as T9 and T10 from the Polish habitat of Zwierzyniec) to 0.758 (between accessions T17 and T21 from the Polish and Norwegian habitats, respectively) (Table S2). In the Norway group, a high degree of genetic similarity was characterized by accessions from Ås, Akershus (T11, T14, T15).

The examined accessions formed two groups on the dendrogram (Figure 1). The first group consisted of accessions from Norway, while the second one grouped Polish accessions. Two classes were distinguished in the Polish group. The first included two genetically indistinguishable accessions from the Bolimów Landscape Park (Budy Grabskie), and the second one included plants from the Zwierzyniec forest complex (Skierniewice). The average value of the genetic diversity indexes of Norwegian accessions compared to the Polish ones were—0.183, 0.170, 0.195 and 0.233 for T11, T14, T15 and T21, respectively. The average value of the genetic diversity indexes of Polish accessions compared to the Norwegian ones were—0.189 for T5, 0.197 for T16 and T20, 0.194 for T19, 0.199 for T18, 0.201 for T12 and T13 and 0.204 for T17 (Table S3). The most genetically diverse accessions were selected for further work towards polyploidisation: originating from Poland (Zwierzyniec)—T16, T20 and T17, and from Norway—T21 and T15 (Figure 1).

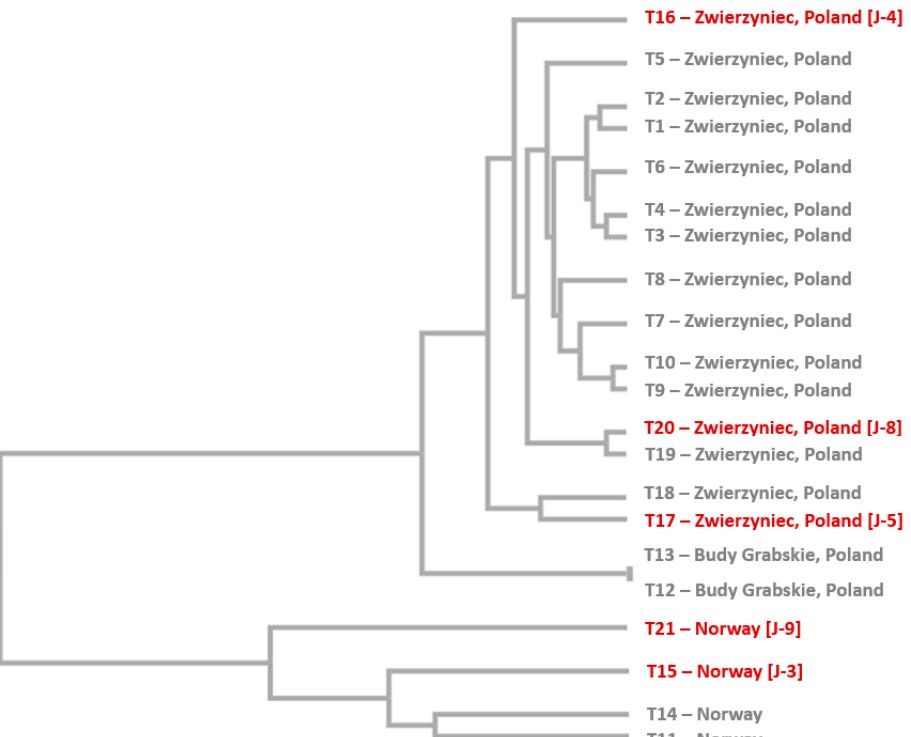

**Figure 1.** The UPGMA dendrogram of 21 bilberry accessions based on AFLP polymorphism. The genotypes selected for polyploidisation are marked in red.

### 3.2. In Vitro Polyploidisation Efficiency

Shoot explants of bilberry accessions J-4, J-8 and J-9 turned out to be more sensitive to the antimitotic agents compared to remaining accessions. Both colchicine and APM caused significant reduction either in the number of explants which survived the antimitotic treatment or the number of regenerated shoots per explant (Table 1, Figure 2A). In accessions J-3 and J-5, antimitotics did not significantly influence the explant survival, but, instead, decreased their regeneration capacity. The highest differences between antimitotic effects were observed in J-9. In this accession, the application of colchicine, irrespective of concentration, and APM at a higher concentration considerably reduced the number of regenerated shoots per explant to 0.8–3.7. APM at a lower concentration of 5 mg $L^{-1}$ resulted in 12 shoots per explant. In the control, the number of regenerated shoots per explant was very high, rearing 34.3.

In total, 995 shoots were subjected to cytometric analysis, with the highest numbers in the accessions J-9 and J-3, and the lowest number in J-4 (Table 2). Tetraploid shoots were selected based on the FCM histograms (Figure 2B). The homogenous tetraploids, 74 in total, were obtained for all of the accessions studied with an average polyploidisation efficiency of 7.1% (Table 2). The highest numbers of tetraploids, 29, 16 and 14, were recorded for J-8, J-3 and J-9, respectively. In the J-4 and J-5 accessions, eight and seven tetraploids were detected, respectively. APM and colchicine, both used at their higher concentrations, revealed a highest polyploidisation efficiency of 10.4% and 8.6%, respectively. The highest number of tetraploid plants was obtained for J-8 (14) following colchicine treatment at 250 mg $L^{-1}$ and for J-3 (11) following APM treatment at 10 mg $L^{-1}$.

The number of mixoploids was 3.5 times higher compared to tetraploids (Table 2). Mixoploids were removed. Selected tetraploids were cultured on multiplication medium to obtain 10–20 shoot clumps and were analysed three times at every subculturing in order to confirm their tetraploidy.

**Table 1.** Effect of antimitotic agents on regenerating capacity of bilberry shoot explants; observed after eight weeks; number of viable explants were calculated per six explants treated with antimitotics.

| Antimitotic (mg $L^{-1}$) | J-3 | J-4 | J-5 | J-8 | J-9 |
|---|---|---|---|---|---|
| Viable explants | | | | | |
| Control | 6.0 a | 6.0 a | 6.0 ab | 6.0 a | 6.0 a |
| Colchicine 125 | 5.8 a | 2.6 b | 5.0 ab | 3.4 b | 2.9 c |
| Colchicine 250 | 5.8 a | 0.8 b | 3.2 b | 4.8 ab | 3.6 bc |
| APM 5 | 5.0 a | 2.5 b | 4.2 ab | 2.8 b | 4.9 ab |
| APM 10 | 5.6 a | 1.4 b | 4.0 ab | 3.2 b | 2.3 c |
| *p* | 0.385 | 0.000 | 0.095 | 0.025 | 0. 000 |
| No. of shoots per explant | | | | | |
| Control | 6.7 a | 10.3 a | 9.7 a | 13.4 a | 34.3 a |
| Colchicine 125 | 3.1 b | 2.1 b | 2.9 bc | 5.0 b | 0.8 c |
| Colchicine 250 | 2.2 b | 0.3 b | 0.6 c | 3.4 b | 3.7 c |
| APM 5 | 6.9 a | 1.2 b | 4.0 b | 2.9 b | 12.0 b |
| APM 10 | 5.1 ab | 0.3 b | 2.7 bc | 2.5 b | 1.7 c |
| *p* | 0.008 | 0.000 | 0.000 | 0.000 | 0.000 |

Mean separation within columns by Duncan's multiple range test. The means (n = 5 there were five replications, i.e., five flasks, each containing six explants) followed by the same letter do not differ at *p* = 0.05; *p*—probability of *F* statistic from one-way ANOVA (separately for each bilberry accessions).

**Table 2.** Numbers of tetraploids obtained from in vitro polyploidisation using shoot explants, analysed using flow cytometry (FCM), 16 weeks after polyploidisation; polyploidisation efficiency was calculated in relation to the number of shoots tested.

| Antimitotic Agents(mg L$^{-1}$) | J-3 | J-4 | J-5 | J-8 | J-9 | Total Number of Tetraploids (Polyploidisation Efficiency, %) |
|---|---|---|---|---|---|---|
| Number of tetraploids (Polyploidisation efficiency, %) | | | | | | |
| Colchicine 125 | 1 (2.7%) | 3 (7%) | 3 (5.8%) | 1 (2.9%) | 6 (12.8%) | 14 (6.6%) |
| Colchicine 250 | 2 (3%) | 0 | 0 | 14 (22.2%) | 5 (10.4%) | 21 (10.4) |
| APM 5 | 2 (3.3%) | 3 (14.3%) | 2 (5.3%) | 8 (16.7%) | 1 (0.7%) | 16 (5.1%) |
| APM 10 | 11 (14.1%) | 2 (28.6%) | 2 (2.3%) | 6 (13.3%) | 2 (3.9%) | 23 (8.6%) |
| Total number of tetraploids, (%) | 16 (6.5%) | 8 (10.1%) | 7 (3.6%) | 29 (15.3%) | 14 (4.9%) | 74 (7.4%) |
| Number of mixoploids (Polyploidisation efficiency, %) | | | | | | |
| Colchicine 125 | 5 (13.5%) | 10 (23.3%) | 7 (13.5%) | 8 (23.5%) | 16 (34.0%) | 46 (21.6%) |
| Colchicine 250 | 7 (10.4%) | 2 (25.0%) | 3 (20.0%) | 14 (22.2%) | 9 (18.8%) | 35 (17.4%) |
| APM 5 | 12 (19.0%) | 13 (61.9%) | 13 (34.2%) | 15 (31.3%) | 31 (21.8%) | 84 (26.0%) |
| APM 10 | 17 (21%) | 5 (71.0%) | 34 (38.6%) | 16 (35.6%) | 21 (41.2%) | 93 (34.6%) |
| Total number of mixoploid (%) | 41 (16.7%) | 30 (38.0%) | 57 (29.5) | 53 (27.9%)) | 77 (26.7%) | 258 (25.9%) |
| Total number of tested shoots | 245 | 79 | 193 | 190 | 288 | 995 |

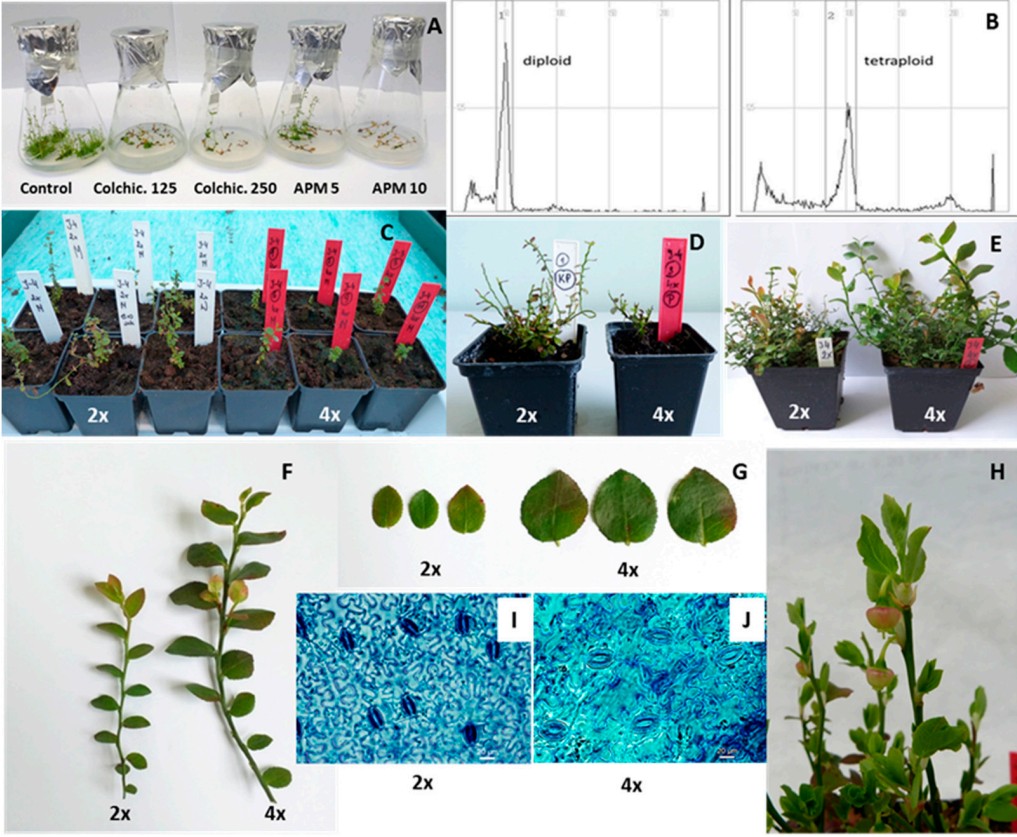

**Figure 2.** (**A**) In vitro induction of bilberry (*Vaccinium myrtillus*) tetraploids using colchicine and amiprophos methyl (APM). (**B**) Histograms of flow cytometry analysis of bilberry regenerated in vitro after antimitotic treatment, from the left: diploid (2x) and tetraploid (4x). Diploid and tetraploid plants grown in greenhouse: (**C**) three-month-old plants (three months after transplanting from in vitro to ex vitro conditions), (**D**) six-month-old plants and (**E**) two-year-old plants. (**F**) Shoots and (**G**) leaves of diploid and tetraploid of bilberry. (**H**) One-and-a-half-year old flowering autotetraploid plant. (**I,J**) Stomata of diploid and autotetraploid.

### 3.3. Phenotypic Evaluation

Tetraploids clearly differed from diploids. The initial growth of tetraploids was much weaker, however, after two years of cultivation, they matched or exceeded the plant vigour of the diploids. The tetraploids were distinguished by significantly larger leaves and higher chlorophyll content, as well as larger stomata and lower stomatal density (Table 3) (Figure 2C–G,I,J). The shoots of tetraploids were generally thicker except for the J-4 4x-1 and 4x-5 clones, which did not differ significantly from their diploid counterpart. Tetraploid clones differ significantly between each other in most parameters, with the exception of stomata density. The largest differences were observed for the relative chlorophyll content in leaves. The highest CCI of 16.7 was observed for the clone 4x-7, while the lowest of 12.3 was found in 4x-1, which did not differ significantly from the diploid with a CCI of 11.4.

**Table 3.** Phenotypic evaluation of the obtained autotetraploid clones of *V. myrtillus* accession J-4 in terms of morphological parameters, the plants were observed after two years of ex vitro cultivation.

| Trait | Diploid | Tetraploids | | | | |
|---|---|---|---|---|---|---|
| | | 4x-1 | 4x-2 | 4x-5 | 4x-7 | *p* |
| Shoot length (cm) | 23.5 ab | 21.4 ab | 24.3 a | 22.1 ab | 20.2 b | 0.17 |
| Shoot thickness (mm) | 2.1 b | 2.4 ab | 2.6 a | 2.1 b | 2.6 a | 0.056 |
| Leaf area (cm$^2$) | 1.6 c | 2.1 bc | 2.4 b | 2.5 b | 3.2 a | 0.000 |
| Chlorophyll content (CCI) | 11.4 d | 12.3 d | 15.4 ab | 13.5 bcd | 16.7 a | 0.001 |
| Stomata length (μm) | 31.6 c | 39.1 a | 39.0 a | 36.5 b | 40.6 a | 0.000 |
| Stomata density (field of view) | 27.5 a | 16.5 b | 16.1 b | 18.3 b | 16.6 b | 0.000 |

Mean separation within rows by Duncan's multiple range test. The means followed by the same letter do not differ at *p* = 0.05; *p*—probability of *F* statistic from one-way ANOVA.

The first flowering of 1.5-year-old plants, either diploid or tetraploid, was observed in April 2021 (Table 4, Figure 2H). Diploids bloomed slightly earlier and slightly more profusely than tetraploid plants. Compared to diploids, the newly obtained autotetraploids had significantly larger flowers by ca. 64% and larger pollen tetrads by ca. 35% (Table 4, Figure 3). The germination capacity of pollen tetrads was high in tetraploids (87.8%), although slightly lower than in diploids (94.3%).

After pollinating the flowers of three highbush blueberry cultivars with pollen from J-4 bilberry autotetraploids, the maternal plants formed fruits (Table 5, Figure 4). The highest percentage of mature fruits, obtained in relation to the number of pollinated flowers, was recorded for 'Bluecrop' (60.7%); less fruits were obtained for 'Liberty' (46.7%) and only 10% for 'Northland'. In total, 9, 11 and 14 seeds were obtained for 'Liberty', 'Bluecrop' and 'Northland', respectively, crossed with tetraploid bilberry J-4. The obtained seeds germinated approximately three weeks after sowing. The effectiveness of interspecific crossing between *V. corymbosum* and tetraploid *V. myrtillus*, defined as the percentage of obtained seedlings in relation to the number of pollinated flowers, was the highest (53.3%) for the blueberry 'Liberty', and 14.8% and 10.0% for 'Bluecrop' and 'Northland', respectively.

**Table 4.** Phenotype evaluation of the diploid and autotetraploid plants of bilberry J-4 accession at flowering phase.

| Trait | Diploid | Tetraploid | *p* |
|---|---|---|---|
| First flower opening time | 20 April | 23 April | - |
| Flowering plants (%) | 66.7 | 20.0 | - |
| Number of flowers/flowering plant | 3.0 | 2.2 | - |
| Flower diameter (mm) | 4.8 b | 7.8 a | 0.008 |
| Pollen tetrad diameter (μm) | 41.9 b | 56.5 a | 0.000 |
| Pollen tetrad germination (%) | 94.7 | 89.3 | - |

Mean separation within rows by Duncan's multiple range test. The means followed by the same letter do not differ at *p* = 0.05; *p*—probability of *F* statistic from one-way ANOVA.

**Table 5.** Efficiency of interspecific crossing between *V. corymbosum* and *V. myrtillus*.

| Parameter | 'Bluecrop' × J-4-4x | 'Northland' × J-4-4x | 'Liberty' × J-4-4x |
|---|---|---|---|
| Number of pollinated flowers | 61 | 70 | 15 |
| Total number of fruits formed (green and matured) | 56 | 11 | 7 |
| % of fruits formed ) in relation to the number of pollinated flowers | 91.8 | 15.7 | 46.7 |
| Number of matured fruits | 37 | 7 | 7 |
| Mature fruit mass (g) | 0.64 | 0.38 | 0.60 |
| Number of well-formed (plump) seeds | 11 | 14 | 9 |
| Number of seedlings obtained | 9 | 7 | 8 |
| % of seedlings in relation to the number of pollinated flowers | 14.8 | 10.0 | 53.3 |

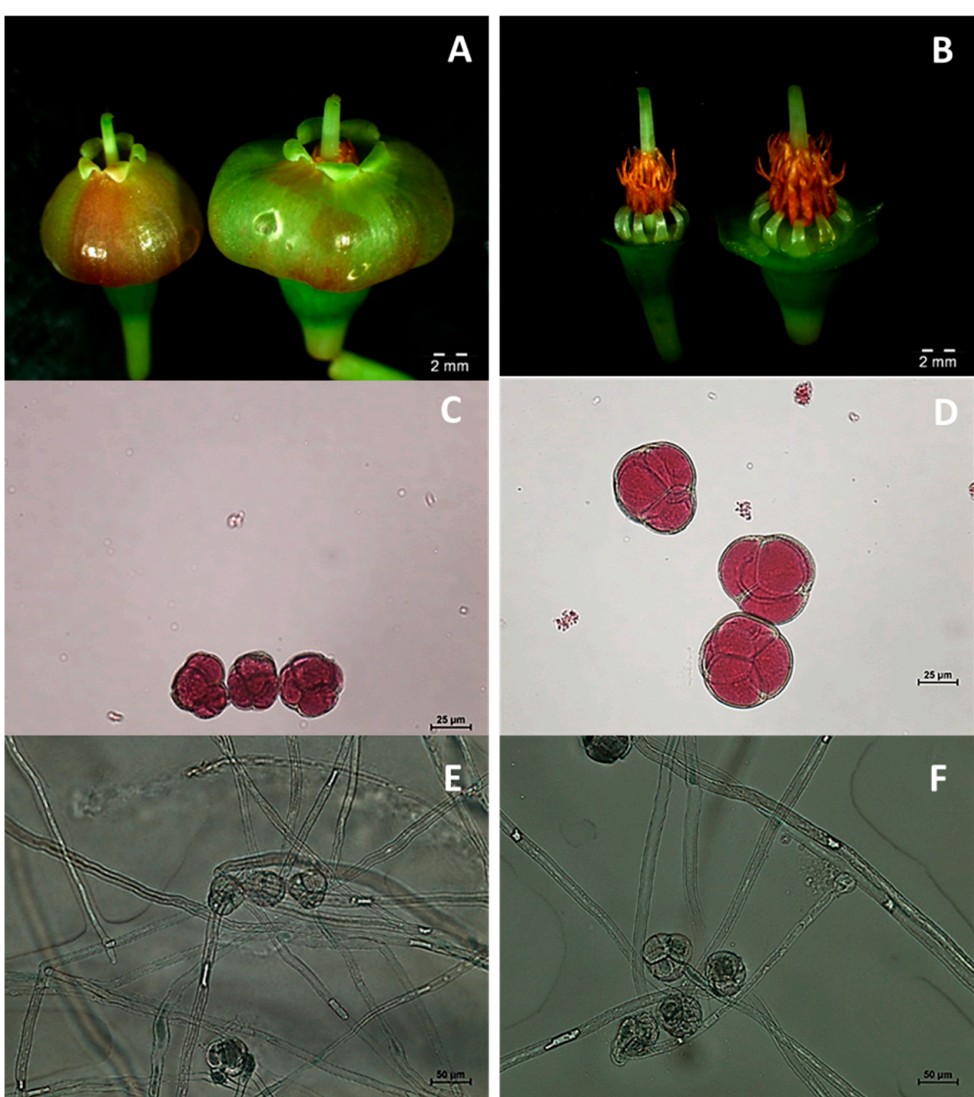

**Figure 3.** (**A**) Flowers and (**B**) flowers without corolla of bilberry J-4 diploid (left) and tetraploid (right). (**C**) Pollen tetrads of diploid and (**D**) tetraploid. Germinating pollen grains (in tetrads) on medium containing sucrose: (**E**) diploid and (**F**) tetraploid.

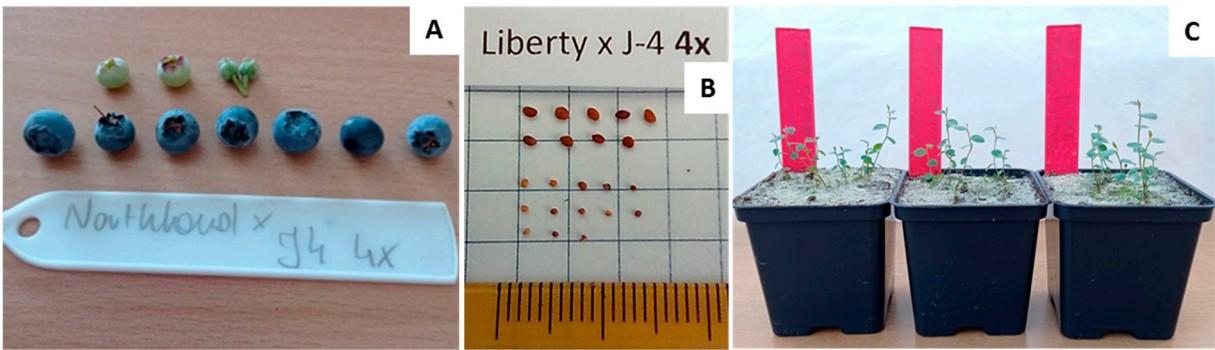

**Figure 4.** (**A**) Fruits, (**B**) seeds and (**C**) seedlings derived from interspecific crosses between maternal plants of *Vaccinium corymbosum* (from the left: 'Bluecrop', 'Northland' and 'Liberty') and paternal plants of *V. myrtillus* autotetraploids.

## 4. Discussion

### 4.1. Bilberry Genetic Variation

Bilberry reproduces vegetatively by rhizomes, and to a lesser extent generatively by seeds, when flowers are pollinated by bumblebees and honeybees. This species is also partially self-pollinating [44]. After self-pollination, fertilisation occurs; however, few seeds are formed [45]. This phenomenon was considered as inbreeding depression or as a mechanism of partial self-incompatibility at the genetic level [20,46]. Due to the prevailing vegetative propagation system and partial selfing, we assumed that genetic variation can be low in *V. myrtillus* within particular populations. Therefore, to increase the pool of genetic diversity for hybridisation with the highbush blueberry cultivars, we selected bilberry accessions for chromosome doubling which were characterised with higher genetic differences based on AFLP analysis.

The examined accessions, gathered in the breeding collection of NIHR, formed two main groups on the dendrogram, with Polish or Norwegian accessions. Results of our study revealed that the genetic similarity indexes between the accessions within the Zwierzyniec population were much higher than those between them and the accessions from the other two populations: from Polish Budy Grabskie and, especially, Norwegian locations. Our results clearly indicated that the increase in genetic differences among the analysed plants was associated with geographical location and distance. Similarly to our studies on accessions derived from the small population of the forest complex Zwierzyniec, Albert et al. [47] analysed the genetic diversity of a small bilberry population using RAPD and AFLP molecular markers. Both molecular markers were effective in bilberry clone identification and the authors reported that genetic distances between pairs of clones were not correlated with the spatial distance between them within this population, which we also observed in our research (e.g., for T5 and T16 obtained from site 1 in Zwierzyniec forest complex). In turn, Zoratti et al. [48] investigated the genetic variation in bilberry within plants collected from different locations in Iceland, Norway, Sweden, Finland and Germany using ISSR markers. The authors revealed the genetic diversity within the studied bilberry populations, indicating the mixed-mating propagation system of bilberry. As in our study, clear correlations were observed between geographic and genetic distances for the populations studied. All these presented results, ours and those of other authors, indicate that, even within a small population, one can find plants that are genetically distinct.

In other studies, several sets of natural *V. myrtillus* populations derived from various locations in the Ardennes in Belgium [49], central Balkans [50], Tuscan Apennines in Italy [51] and Baltic countries (Latvia, Lithuania and Estonia) [52] were also assessed in the context of genetic variability. The results of these extensive studies confirmed the observations in our study and that of Zoratti et al. [48] which indicated that, in bilberry, the intra-population genetic variation was moderate to significant, and it was very high between populations derived from remote regions.

Similar patterns of genetic variability within a population and among geographically distant populations were reported for other *Vaccinium* species. In Lithuania, Žukauskienė et al. [53], using RAPD analysis, showed a high level of genetic variation among cranberry (*V. oxycoccus*) populations, whereas low variability was found within populations. Rodriguez-Bonilla et al. [54], analysing populations of *V. macrocarpon* and of *V. oxycoccus*, using SSR markers, observed similar clear geographical separation among the populations of both species from western, central and eastern locations of the US National Forests. Recently, Vega-Polo et al. [55] reported a high degree of genetic diversity for Andean blueberry (*Vaccinium floribundum* Kunth.) populations in the Ecuadorian highlands. Based on SSR markers, these authors revealed the existence of distinct genetic clusters present in the northern, central and southern highlands with sub-clusters clearly differentiated from locations at higher elevations. In contrast, Debnath [28] reported that a wide genetic diversity was found, using ISSR markers, among the clones of *V. angustifolium*; however, no pattern of geographical differentiation was shown.

Our results indicate that the selection of genotypes in order to induce bilberry tetraploids, which are then to be crossed with highbush blueberry, should be preceded by an analysis of genetic variation. Such a procedure, in conjunction with observations of both the original diploids and the autotetraploids obtained from them, will allow for the selection of the most valuable individuals for future crossings.

It should be noted that our criteria for selecting appropriate genotypes for polyploidisation were not only genetic differences, but also plant vigour, flowering and fruiting in cultivation conditions which are other than natural. Our bilberry accessions were grown in containers in a substrate similar to that used for highbush blueberry, with the addition of forest soil, in a greenhouse. For in vitro shoot culture initiation (needed for in vitro polyploidisation), eight accessions were selected, being, firstly, those which grew best under these unnatural conditions and, at the same time, were characterised by a high degree of genetic diversity. However, the three genotypes showed poor shoot formation ability in in vitro cultures. Therefore, subsequently, five genotypes with higher regeneration capacity were chosen for polyploidisation.

### 4.2. Bilberry Polyploidisation and Evaluation of Autotetraploids in Terms of Phenotype and Cross Ability

The efficiency of bilberry polyploidisation achieved in our research was relatively high. A single series of antimitotic treatments for each accession was sufficient to obtain from 7 to 29 tetraploids for all of them. Colchicine and APM at the higher concentrations of 250 mg L$^{-1}$ and 10 mg L$^{-1}$, respectively, showed a similar tetraploidy-induction efficiency of about 9–10%. In earlier studies on polyploidisation of daylily and apple, oryzalin and trifluralin were used in addition to colchicine and APM. However, as in the case of bilberry, the greatest numbers of tetraploids were obtained after treatments with colchicine and APM [14,15]. APM, being less toxic to humans, seems to be a good alternative to colchicine for chromosome doubling. This antimitotic was applied successfully for genome duplication in numerous plant species in order to produce dihaploids, e.g., of onion [56] or maize [57] and, as recently reported, to obtain tetraploids of *Dendrobium* [58] and *Antirrhinum majus* [59].

Previously, mitotic tetraploids were induced mainly by soaking seeds in colchicine solutions. Such tetraploids were produced for a few diploid species: *V. arboretum* (*Batodendron* section), *V. fuscatum* (*Cyanococcus* section) and *V. stamineum* (*Polycodium* section) [8,12,13]. These authors then obtained hybrids between *V. corymbosum* and autotetraploids of the above-mentioned three species. Similarly, Tsuda et al. [60] produced intersectional hybrids between highbush blueberry 'Spartan' and colchicine-induced tetraploid of *V. bracteatum*— a wild diploid species native to Japan, whose fruits are characterised by anthocyanin-containing flesh. Tsuda et al. [60] induced chromosome doubling in this *Vaccinium* species via in the vitro culturing of seeds on colchicine-containing medium. These authors obtained several hybrids, and some of them had berries with violet or semi-violet fruit flesh.

However, to the best of our knowledge, we have not found any reports on hybrids between bilberry and blueberry.

Only one report was found on the induction of tetraploids within *Vaccinium* using non-seed explants [61]. The authors of the mentioned study obtained a tetraploid clone of *V. elliottii* by in vitro treatment with colchicine of two-node cuttings. This tetraploid clone was then evaluated for morphological features and the ability to cross with *V. corymbosum* [62]. Similarly to our bilberry tetraploids, the *V. elliottii* tetraploids had thicker shoots, larger flowers, leaves that were twice as large, stomata that were 40% longer and larger pollen grains compared to wild *V. elliottii*. The viability of tetraploid pollen (27.7%), assessed by germination on a sucrose containing medium, was, however, significantly lower than that of a diploid (53.3%). Despite this, the tetraploid clone was characterised by relatively high fertility when crossing with *V. corymbosum*; in crossing, when the tetraploid of *V. elliottii* occurred as a seed-plant, 7.2 seedlings per pollinated flower were obtained, and, in the reciprocal crossing, two times less seedlings were obtained.

In the literature, one report was found referring to bilberry autotetraploids; only two tetraploid plants were obtained years ago which died, although they had survived for two years [63]. The vigorous bilberry autotetraploids we obtained are probably the first such plants.

Another approach for overcoming the barrier of interploid crosses in *Vaccinium* species is to use genotypes capable of producing unreduced gametes. Several valuable meiotic tetraploid hybrid seedlings were produced from crossing *V. corymbosum* with diploid *V. elliotti* [10]. Such meiotic tetraploids were fertile and vigorous, and the fruit size was superior to that of the wild diploid *V. elliotti*. In a multi-stage breeding process (F1, F2 and backcrosses to highbush blueberry), Cabezas et al. [10] obtained a number of interesting hybrids with commercially acceptable fruit size similar to that of highbush blueberry. However, obtaining meiotic interspecific hybrids from interploid crosses is very difficult. Norden et al. [9] obtained only 18 triploid and 3 tetraploid hybrid seedlings by pollinating more than four thousand flowers of *V. corymbosum* with the pollen of diploid *V. elliotti* from 19 plants of this species. All tetraploid hybrids were fertile and had larger organs, including fruits, compared to wild *V. elliotti*, and were incorporated into further breeding. In contrast, all triploids were completely sterile. These authors also obtained a few tetraploid hybrid seedlings from reciprocal crosses. Unfortunately, the crossing of several cultivars of highbush blueberry with wild bilberry carried out at NIHR failed (data not presented). Thus far, no reports on obtaining such hybrids have been found in the literature.

Bilberry autotetraploids obtained in our research are characterised by high vigour and viable pollen. We also obtained seedlings from crosses between three cultivars of *V. corymbosum* as a seed parent and autotetraploids of *V. myrtillus* as a pollen parent. Before using them to breed improved highbush blueberry cultivars, it is planned to confirm the hybridity of the seedlings using molecular markers and to evaluate their phenotypes.

**Supplementary Materials:** The following are available online at https://www.mdpi.com/article/10.3390/agronomy11122584/s1, Figure S1: Characteristics of amplified fragment length polymorphism (AFLP) primers used to assess the genetic diversity of bilberry (*Vaccinium myrtillus*) accessions, Table S1: Characteristics of amplified fragment length polymorphism (AFLP) primers used to assess the genetic diversity of bilberry (*Vaccinium myrtillus*) accessions, Table S2: The similarity index matrix of bilberry accessions based on AFLP markers, Table S3: The genetic distance index matrix of bilberry accessions based on AFLP markers.

**Author Contributions:** Conceptualization, M.P., S.P., K.M.; Funding acquisition, M.P.; Investigation, M.P., K.M., M.M., A.M.-C., S.P.; Methodology, M.P., M.M., A.M.-C.; Project administration, M.P.; Resources, M.P.; Supervision, M.P.; Writing—Original draft, M.P., K.M., M.M., S.P., A.M.-C.; Writing—Review & editing, M.P. All authors have read and agreed to the published version of the manuscript.

**Funding:** The study was funded by the Polish Ministry of Agriculture and Rural Development, Biological Progress, Task No. 45 (2021).

**Institutional Review Board Statement:** Not applicable.

**Informed Consent Statement:** Not applicable.

**Data Availability Statement:** All data were presented in this paper.

**Conflicts of Interest:** The authors declare no conflict of interest.

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
