# Peer review of "Chromosome Doubling in Genetically Diverse Bilberry (Vaccinium myrtillus L.) Accessions and Evaluation of Tetraploids in Terms of Phenotype and Ability to Cross with Highbush Blueberry (V. corymbosum L.)"

_agronomy, doi:10.3390/agronomy11122584_

Round 1

Reviewer 1 Report

The English language is generally good. There are, however, tens of hyphenations which shall be removed. Al the scientific names must be written in italics.

Line 134: lowbush: replace: lowbush blueberry

Section 2:

2.1: Give the country

Give the years of treatments

Line 198: Give more accurate information of LED lamps

Line 206: Explain if colchicine and APM were added before autoclaving or sterile-filtered after autoclaving

Line 214: Explain more accurately, what you mean with “random system”.

Lines 217-218: Too many parentheses, remove the latter ones.

Lines 236-237, Check the degree marks to be correctly printed (as seen on line 197). in October: give the year.

Line 240: Check the Litre mark

Lines 245-248. Explain more carefully, if J4 plants were from some previous experiment , or from this study.

Line 277: April, give the year

Line 338-340: Improve the language.

Line 354-355: Improve language, do you mean highest number of tetraploid plants?

Results:

In Results: Do not repeat values that you give in the tables! Especially several %-values are now repeated.

Lines 366-370. The sentence is difficult to understand, split it into two parts.

In Results, also respect the results of Duncan’s testing, when you report the results. Explain, when you refer to the results of ANOVA on general level, but do not omit Duncan-s results, because there seam to be differences between clones (for instance on lines 368-369).

When giving percentages, decimals are not needed (%s are already hundredth parts).

Discussion:

The discussion is very long and shall be reduced. Avoid reporting the results of other scientists’ previous research in details, because this is not a review article.

The first paragraph of discussion is far too long. Divide it to several parts. It also repeats some info, remove all unnecessary and write in shorter way, especially lines 426-440.

Lines 421-422: The sentence must be modified (Due to …)

Line 451: genetic bilberry variation:  do you mean bilberry genetic variation?

References (numbers refer to those given by authors):

Remove the linebreak in Ref. 2 within the title name.

Check the italics (in several refs.) and hyphens (at least in Refs. 13, 14, 26, 36). Check the style /info of Refs., especially the following: 1, 5 [URL (accessed on Day Month Year)], 10, 11, 24, 26-27?, 34 (one page review?), 48,  49, 54

Tables:

Use decimal point and not comma (Tables 3, 5). use either capital P or lower case p, not both. Give explanation that P refers to ANOVA.

Use 3 decimal places to P, and use P<0.001 when necessary in Tables 3, 5, 6

Table 3:Check the presentation of Duncan’s test results for Viable explants of J-5 (also consider those of J-3). In Table 3, give the number of explants in the explanatory title, not connected to “Viable explants”. Explain what is n=5 in the subtitle.

Table 4: Check table title (only tetraploids mentioned), Improve the language of subtitles (Tetraploid number, Mixoploid number, preferably Number of…)

Table 7: Improve accuracy of line titles: Mature fruits % (of what?), Number of well-formed seeds % (number --%???, Of what?)

Figure 3 title. Start the sentence with capital letter in C.

Author Response

Dear Reviewers,

We are very grateful to the reviewers for very careful reviews, for their suggestions and for any bugs found. Thanks to their work, we were able to improve the quality of our manuscript. Many thanks.

Reviewer 1

The English language is generally good. There are, however, tens of hyphenations which shall be removed. Al the scientific names must be written in italics.

Response: All hyphenations were removed and the Latin plant names have been written in Italic; superscripts, and subscripts were applied for grades etc.. These mistakes occurred probably following the automatic MDPI formatting, similarly as lack of superscripts, and subscripts, and the additional numbering of references.

These mistakes were corrected without using the function “Track changes”. All other changes were done using “track changes” function.

Line 134: lowbush: replace: lowbush blueberry.

Response: it has been corrected

Section 2:

2.1: Give the country

Response: “Poland” has been added

Give the years of treatments

Response: In preliminary study, in 2018, in vitro shoot cultures were initiated and polyploidization was done for the bilberry accession J-4. The J-4 tetraploids were then rooted, cultivated ex vitro during 2019- 2021, and finally their phenotype was observed in 2021. In the case of other genotypes: J-3, J-5, J-8 and J-9, in vitro shoot cultures were established at the beginning of 2021 and soon the obtained shoots were polyploidized. In vitro cultures and polyploidization were performed for all five genotypes according to the same procedures described below. It was explaned in text 2.1.

Line 198: Give more accurate information of LED lamps

Response: more information were added in the manuscript.

Line 206: Explain if colchicine and APM were added before autoclaving or sterile-filtered after autoclaving

Response:  Colchicine was dissolved in distilled water and filtered through microporous filter membrane of 0.22 μm (NalgeneTM Sterile Syringe Filters, Thermo Fisher Scientific, Waltham, MA, USA), and APM was dissolved in 99.8% ethanol and directly added to the medium. The antimitotics were included to the media after autoclaving.

Line 214: Explain more accurately, what you mean with “random system”.

Response: This sentence was removed. There were 5 flasks each containing 6 explants. The flasks were placed randomly on the shelf in the growth room. (Line: 226-228)

Lines 217-218: Too many parentheses, remove the latter ones.

Response: it was corrected.

Lines 236-237, Check the degree marks to be correctly printed (as seen on line 197). in October: give the year.

Response: It was corrected

Line 240: Check the Litre mark

Response: It was corrected

Lines 245-248. Explain more carefully, if J4 plants were from some previous experiment , or from this study.

Response: We have explained it above.

Line 277: April, give the year.

Response: It was corrected: In  2021.

Line 338-340: Improve the language.

Response: We tried to correct it.

Line 354-355: Improve language, do you mean highest number of tetraploid plants?

Response: Yes, it was corrected according to reviewer suggestion.

Results:

In Results: Do not repeat values that you give in the tables! Especially several %-values are now repeated.

Author’s response: The 'Results' section was corrected according to suggestion.

Lines 366-370. The sentence is difficult to understand, split it into two parts.

Response: Corrected.

In Results, also respect the results of Duncan’s testing, when you report the results. Explain, when you refer to the results of ANOVA on general level, but do not omit Duncan-s results, because there seam to be differences between clones (for instance on lines 368-369).

Response: We have detailed the description of the differences between the tetraploid clones.

When giving percentages, decimals are not needed (%s are already hundredth parts).

Response: This was corrected in Results 3.1.

Discussion:

The discussion is very long and shall be reduced. Avoid reporting the results of other scientists’ previous research in details, because this is not a review article.

The first paragraph of discussion is far too long. Divide it to several parts. It also repeats some info, remove all unnecessary and write in shorter way, especially lines 426-440.

Lines 421-422: The sentence must be modified (Due to …)

Line 451: genetic bilberry variation:  do you mean bilberry genetic variation?

Author’s response: The first part of ‘Discussion’ was corrected according to reviewer’s suggestion. Text in lines 421-422 and 451 was also corrected.

References (numbers refer to those given by authors):

Remove the linebreak in Ref. 2 within the title name.

Check the italics (in several refs.) and hyphens (at least in Refs. 13, 14, 26, 36). Check the style /info of Refs., especially the following: 1, 5 [URL (accessed on Day Month Year)], 10, 11, 24, 26-27?, 34 (one page review?), 48,  49, 54

Response: We corrected these mistakes.  In the case of some journals (e.g. these of MDPI) which are published only on line, there are no numbers of pages given; only the number of the article is given. 

Tables:

Use decimal point and not comma (Tables 3, 5). use either capital P or lower case p, not both. Give explanation that P refers to ANOVA.

Use 3 decimal places to P, and use P<0.001 when necessary in Tables 3, 5, 6

Response: We corrected our mistakes and added the explanation for “p” of ANOVA under the table 1(3 before revision), 2(4) and 4(6)

Table 3 (now table 1):Check the presentation of Duncan’s test results for Viable explants of J-5 (also consider those of J-3). In Table 3, give the number of explants in the explanatory title, not connected to “Viable explants”. Explain what is n=5 in the subtitle.

Response: Duncan test is proper. Difference between means are not significant (high SD).

There were 5 flasks each containing 6 explants. The flasks were placed randomly on the shelf in the growth room.

Table 4 (now table 2): Check table title (only tetraploids mentioned), Improve the language of subtitles (Tetraploid number, Mixoploid number, preferably Number of…)

Table 7 (now table 5): Improve accuracy of line titles: Mature fruits % (of what?), Number of well-formed seeds % (number --%???, Of what?)

Response: Descriptions in the tables have been corrected and supplemented as suggested by the reviewer.

Figure 3 title. Start the sentence with capital letter in C.

Response: corrected

Reviewer 2 Report

This manuscript presents the results of an interesting research on a relevant topic, fully corresponds to the journal area.

There are some comments and suggestions which have to be answered.

Line 217. Please check the reference [15], perhaps it is incorrect.

Line 245. It is not clear what the authors mean under “…the first and the earliest series of polyploidisation”. Please, make this phrase more comprehensive.

Line 258. Please check the reference [42] which denotes Dyki, B.; Habdas, H. The method of isolation of epidermis of tomato and cucumber leaves for microscopic in-vestigation of pathogenic fungus development. Acta Agrobot. 1996, 49, 123-129. If this publication for the methods to access size and density of stomata indeed?

Line 325. Values of the genetic diversity out to be noted on the dendrogram

Line 329 and 331. Table 1 and Table 2 will be more appropriate as the Supplement, probably.

Line 362. Table 4. Total number of mixoploids obtained after Colchicine 250 treatment is incorrect obviously. Due to the sum of five figures it is 35, not 25.

Line 421. “Due to the prevailing vegetative propagation system and partial selfing.” This phrase need to be revised.

Line 427. “Each clone was formed by closely packed ramets belonging to the same genet.” This phrase need to be revised.

Author Response

Dear Reviewer,

We are very grateful to the reviewers for very careful reviews, for their suggestions and for any bugs found. Thanks to their work, we were able to improve the quality of our manuscript. Many thanks.

The English language is generally good. There are, however, tens of hyphenations which shall be removed. Al the scientific names must be written in italics.

Response: All hyphenations were removed and the Latin plant names have been written in Italic; superscripts, and subscripts were applied for grades etc.. These mistakes occurred probably following the automatic MDPI formatting, similarly as lack of superscripts, and subscripts, and the additional numbering of references.

These mistakes were corrected without using the function “Track changes”. All other changes were done using “track changes” function.

Reviewer 2

This manuscript presents the results of an interesting research on a relevant topic, fully corresponds to the journal area.

There are some comments and suggestions which have to be answered.

Line 217. Please check the reference [15], perhaps it is incorrect.

Response: We removed it,. The proper reference is 41 (Śliwińska, 2008)

Line 245. It is not clear what the authors mean under “…the first and the earliest series of polyploidisation”. Please, make this phrase more comprehensive.

Response: We explained it in MM 2.1.

Line 258. Please check the reference [42] which denotes Dyki, B.; Habdas, H. The method of isolation of epidermis of tomato and cucumber leaves for microscopic in-vestigation of pathogenic fungus development. Acta Agrobot. 1996, 49, 123-129. If this publication for the methods to access size and density of stomata indeed?

Response: The procedure for stomata measurements is described in this paper (the article is available in the internet),  https://pbsociety.org.pl/journals/index.php/aa/article/view/aa.1996.013/1975

Line 325. Values of the genetic diversity out to be noted on the dendrogram

Author’s response: The program for preparing the dendrogram does not place the scale with the genetic diversity index on it. Therefore, tables with the values of the genetic diversity and similarity indexes were attached to the manuscript (supplementary materials).

Line 329 and 331. Table 1 and Table 2 will be more appropriate as the Supplement, probably.

Author’s response: We agree with this suggestion. Tables 1 and 2 have been moved to Supplementary materials.

Line 362. Table 4. Total number of mixoploids obtained after Colchicine 250 treatment is incorrect obviously. Due to the sum of five figures it is 35, not 25.

Response: our mistake was corrected

Line 421. “Due to the prevailing vegetative propagation system and partial selfing.” This phrase need to be revised.

Response: The sentence was corrected.

Line 427. “Each clone was formed by closely packed ramets belonging to the same genet.” This phrase need to be revised.

Response: This sentence was removed. In general the discussion was reduced.